# Peripheral and Central Iron Measures in Alcohol Use Disorder and Aging: A Quantitative Susceptibility Mapping Pilot Study

**DOI:** 10.3390/ijms24054461

**Published:** 2023-02-24

**Authors:** Aiden R. Adams, Xinyi Li, Juliana I. Byanyima, Sianneh A. Vesslee, Thanh D. Nguyen, Yi Wang, Brianna Moon, Timothy Pond, Henry R. Kranzler, Walter R. Witschey, Zhenhao Shi, Corinde E. Wiers

**Affiliations:** 1Center for Studies of Addiction, Department of Psychiatry, Perelman School of Medicine, University of Pennsylvania, 3535 Market St Ste 500, Philadelphia, PA 19104, USA; 2Department of Radiology, Weill Cornell Medicine, 525 E 68th St, New York, NY 10065, USA; 3Department of Radiology, Perelman School of Medicine, University of Pennsylvania, 3400 Civic Center Blvd, South Pavilion, Room 11-155, Philadelphia, PA 19104, USA

**Keywords:** iron, ferritin, alcohol use disorder, quantitative susceptibility mapping

## Abstract

Chronic excessive alcohol use has neurotoxic effects, which may contribute to cognitive decline and the risk of early-onset dementia. Elevated peripheral iron levels have been reported in individuals with alcohol use disorder (AUD), but its association with brain iron loading has not been explored. We evaluated whether (1) serum and brain iron loading are higher in individuals with AUD than non-dependent healthy controls and (2) serum and brain iron loading increase with age. A fasting serum iron panel was obtained and a magnetic resonance imaging scan with quantitative susceptibility mapping (QSM) was used to quantify brain iron concentrations. Although serum ferritin levels were higher in the AUD group than in controls, whole-brain iron susceptibility did not differ between groups. Voxel-wise QSM analyses revealed higher susceptibility in a cluster in the left globus pallidus in individuals with AUD than controls. Whole-brain iron increased with age and voxel-wise QSM indicated higher susceptibility with age in various brain areas including the basal ganglia. This is the first study to analyze both serum and brain iron loading in individuals with AUD. Larger studies are needed to examine the effects of alcohol use on iron loading and its associations with alcohol use severity, structural and functional brain changes, and alcohol-induced cognitive impairments.

## 1. Introduction

Iron is an essential cofactor for many cellular enzymes involved in metabolic pathways including DNA synthesis, oxygen transport in the blood, and neurotransmitter synthesis [1,2]. However, iron can also promote oxidative damage by catalyzing free radical reactions that denature DNA and proteins [3]. Excess iron deposition in the brain plays a significant role in the pathology of neurodegenerative diseases, such as Alzheimer’s disease (AD), Parkinson’s disease (PD), and Huntington’s disease (HD) [4,5]. Serum and brain iron loading have also been reported in individuals with alcohol use disorder (AUD) [6], but its impact on the clinical course of the disease is not well understood.

Peripheral iron status is monitored clinically with serum iron level, percent saturation of the iron transporter protein transferrin, and iron storage protein ferritin. We recently reported that serum transferrin saturation was higher in individuals with AUD than non-dependent healthy controls and positively correlated with the severity of alcohol withdrawal [7]. Excessive alcohol consumption was also linked to increased levels of peripheral ferritin, which may contribute to the toxic effects of alcohol on tissues, including the liver and brain [8]. As such, ferritin levels increased with the severity of liver disease [8]. Brain iron concentrations can be quantified by magnetic resonance imaging (MRI) quantitative susceptibility mapping (QSM), which infers local tissue susceptibility secondary to the presence of iron and other minerals [9,10,11]. One previous study demonstrated higher brain iron levels in the basal ganglia in AUD participants than healthy controls through retrospective functional MRI (fMRI) QSM analysis [6]. The means by which iron accumulates in the brain in the context of AUD is not well understood, but one theory postulates that chronic neuroinflammation from AUD leads to a “leakier” blood brain barrier (BBB) [12]. This may allow iron to cross the BBB more easily and accumulate in brain tissues.

Iron deposition in individuals with AUD may contribute to neurodegeneration similar to that observed with aging and neurodegenerative diseases, such as AD and PD [13]. Peripheral iron levels are abnormally high early in the course of AD and PD [14]. QSM analysis of brain iron corroborated the serum findings showing brain iron accumulation in brain regions associated with motor and cognitive impairment in patients with AD and PD [15]. Higher iron accumulation was positively associated with positron emission tomography (PET) measures of amyloid and tau, biomarkers of AD, in the basal ganglia, suggesting that high magnetic susceptibility in certain deep gray nuclei may reflect iron’s role as a marker of cognitive decline [16]. Sequestration of iron through dietary modulation or iron chelation was effective in attenuating brain iron build-up and cognitive decline [17].

Brain iron loading has also been implicated in normal aging. Results from both early post-mortem studies and recent in vivo QSM studies indicated that elderly subjects had significantly higher brain iron concentrations than younger ones in various subcortical regions including the putamen and globus pallidus [18,19]. Other MRI studies have revealed significant age-related iron differences in cortical gray matter and motor/premotor cortices using both traditional R2* (transverse relaxation-dephasing rate) and the more novel QSM method [20]. A recent study also showed that abnormal function of the glymphatic system, which is responsible for waste clearance in the central nervous system, may be associated with age-related brain iron deposition [21].

Few studies have examined the role of iron in mediating AUD- and aging-associated neurodegeneration. In this study, we utilize QSM to (1) investigate serum and brain iron loading in individuals with AUD and (2) examine the effects of age on serum and brain iron levels. We hypothesized elevated blood ferritin and brain levels of iron in individuals with AUD compared with controls and significant effects of age on both measures. We also performed a re-analysis of iron panel data from AUD participants (*n* = 583) and controls (*n* = 470) from the National Institutes of Health (NIH) Clinical Center (Unit and Clinic Evaluation, Screening, Assessment, and Management) [7] to assess serum ferritin levels.

## 2. Results

### 2.1. Higher Serum Ferritin Level and Concentration in Individuals with AUD Than Controls

Data from the NIH Clinical Center showed the AUD group to have higher serum ferritin levels (mean ± SD = 252.4 ± 282.0 *n* = 583) than the HC group (91.0 ± 112.1; t_794.7_ = 12.6, *n* = 470, *p* < 0.0001) (Figure 1A). A linear regression model demonstrated that AUD status was a significant predictor of serum ferritin level (*β =* 0.256, *p* < 0.001) after controlling for age, sex, BMI, and C-reactive protein (CRP) levels. Age (41.3 ± 12.9, range = 18–77 years old) was also a predictor of serum ferritin level (*β =* 0.080, *p* = 0.002) (Appendix A).

Demographic and clinical characteristics of the AUD (*n* = 10) and healthy control (*n* = 8) cohorts from the University of Pennsylvania (Penn) sample, who underwent both laboratory testing and neuroimaging with QSM, are summarized in Table 1. Participants were a mean of 35.2 ± 12.9 years old (range = 22–62). The results from Penn also indicated that serum ferritin levels were higher in individuals with AUD than healthy controls (t_16_ = 2.5, *p* = 0.023) (Figure 1B). A linear regression model also indicated that AUD status was a significant predictor of ferritin (*β =* 0.523, *p* = 0.049) when controlling for age, sex, and BMI (Appendix A). However, age did not significantly predict serum ferritin in the Penn sample (*β* = −0.325, *p* = 0.14). Group means and comparisons of other iron panel measures are listed in Appendix A.

### 2.2. Associations of AUD and Age with Global Brain Iron Susceptibility

Penn participants underwent a 3 Tesla MRI scan with QSM to non-invasively quantify brain iron concentrations. QSM analysis was performed using the MEDI toolbox and global brain susceptibility was extracted (see Section 4). No outliers (>2SD from the mean) were detected. Linear regression analysis demonstrated that age (*β =* 0.618, *p* = 0.020) and sex (*β* = −0.491, *p* = 0.047) significantly predicted global brain susceptibility (Figure 2 and Table 2), with older age and male sex predicting higher global brain susceptibility. However, neither AUD (*β* = −0.775, *p* = 0.45) nor serum ferritin (*β =* 0.357, *p* = 0.21) significantly predicted global brain susceptibility (Table 2).

### 2.3. Whole-Brain Voxel-Wise QSM Analysis

There were no significant associations of age or AUD with brain quantitative susceptibility using a family-wise error (FWE)-corrected *p*-value (P_FWE_ < 0.05). Exploratory analyses with a more liberal threshold of voxel-level uncorrected-*p* < 0.005 combined with cluster extent *k* > 50 demonstrated higher brain quantitative susceptibility in the left globus pallidus in AUD individuals than controls (peak voxel coordinates = [−26, −21, 6], *k* = 89, Z = 2.90, uncorrected-*p* = 0.002; Figure 3A). Moreover, age was positively correlated with brain quantitative susceptibility in the precuneus, bilateral dorsolateral prefrontal cortex (dlPFC), orbitofrontal cortex (OFC), bilateral midbrain, bilateral fusiform gyrus/parahippocampal gyrus, left thalamus, and right caudate (all at voxel-level uncorrected-*p* < 0.005, *k* > 50, Figure 3B and Appendix A).

## 3. Discussion

We found significantly higher serum ferritin levels in the AUD group than in healthy controls in two independent cohorts (NIH and Penn). Although whole-brain quantitative susceptibility did not differ between the AUD and control group, a preliminary voxel-wise QSM analysis revealed higher brain susceptibility in the left globus pallidus in AUD individuals than in healthy controls. Age correlated positively with whole-brain iron accumulation and further voxel-wise analysis showed correlations in the precuneus, bilateral dlPFC, OFC, bilateral midbrain, precentral gyrus, fusiform gyrus, left thalamus, and right caudate.

Our findings of higher serum ferritin and greater globus pallidus brain iron in AUD are in line with that reported in previous studies [6,7,22]. Evidence suggests that the globus pallidus is involved in inhibitory control and a positive association was found between globus pallidus functional connectivity and clinical measures of impulsivity (i.e., Barratt Impulsiveness Scale and UPPS-P Impulsive Scale) [23]. Impulsivity has been associated with an increased risk of problematic drinking behavior and AUD [24]. Therefore, iron loading in the globus pallidus could contribute to altered neuronal functioning and the development of AUD. Serum ferritin is widely accepted as a sensitive marker of systemic iron stores and ferritin is commonly used in clinical practice as a diagnostic test for iron deficiency and overload [25,26]. Low serum or plasma ferritin levels indicate iron deficiency, while elevated ferritin levels reflect potential iron overload [25,26,27]. Elevated peripheral iron measures, including serum ferritin and transferrin saturation, have been associated with brain iron stores in published human neuroimaging studies [28,29]. Serum ferritin is limited as a clinical marker of total iron body stores insofar as it is an acute-phase protein that is elevated by inflammation and infection, including HIV, pregnancy, and late-stage kidney disease [26]. Despite having excluded participants with these conditions from our sample, alcohol-induced chronic inflammation may have elevated ferritin and brain iron levels in the AUD group. Kroll et al. found elevated levels of CRP, a marker of inflammation, in the AUD group compared with controls [7]. In the NIH sample, CRP levels were positively associated with serum ferritin levels (see Appendix A). However, controlling for CRP in the regression model did not influence the significant effects of AUD on serum ferritin levels (Appendix A). While chronic hepatitis has been associated with elevated brain iron deposition [30], hepatitis likely did not play a role in this neuroimaging study because individuals with abnormal liver enzyme panel results and/or liver disease were excluded.

We found no significant group differences in brain iron at the whole-brain level. Previous QSM studies showed increased brain iron in the globus pallidus, as well as the caudate nucleus, dentate nucleus, putamen, and substantia nigra in AUD individuals compared with controls [6,22]. Discrepancies in study findings may be attributable to our small sample size (i.e., *n* = 18 participants in our study versus *n* = 35 in [6] and *n* = 20,729 U.K. Biobank participants in [22]), and differences in the acquisition of QSM. Our study acquired detailed, four-echo three-dimensional images of tissue magnetic susceptibility, which allow for more accurate and precise measurement than other MRI techniques. Previous studies utilized T2*-weighted imaging of only one or two echoes [6,22]. Furthermore, the greater brain iron reported by Juhás et al. [6] was obtained in male inpatients with AUD, whereas our AUD population included non-treatment-seeking males and females. This is particularly relevant because male sex was associated with higher peripheral ferritin levels and global brain iron levels in our study. Topiwala et al. [22] found an association between alcohol consumption and striatal (putamen and caudate) iron levels, an association mediated by liver iron levels. However, we found no association between serum ferritin and total brain iron, suggesting that there is regulation of brain iron by the BBB and iron transporter proteins [31,32]. The link between blood and brain iron levels requires further elucidation as the physiological mechanism behind iron loading in humans is not well studied. Exploring the relationship among serum, liver, and brain iron measures could enhance our understanding of the disease progression of AUD [22].

The age-associated brain iron accumulation observed in our studies is consistent with that found in prior QSM studies [20,33]. Excessive iron deposition elicits oxidative stress, which facilitates neurodegeneration and impairs neuronal function [34]. Although the mechanisms underlying brain iron accumulation remain largely unknown, it appears to be region-specific and some brain regions are more efficient in maintaining iron homeostasis than others [34]. Here, we found significant associations between aging and iron in brain regions involved in motor and cognitive functioning [35,36], in line with previous studies that associated brain iron concentration with age-related cognitive and motor decline [37,38,39]. The variations in brain regions affected by age-related iron loading reported here have also been observed in other studies [20,33]. These differences may be attributable to study differences in QSM analysis, the age distribution of participants, or the severity of AUD, all of which require elucidation. Our study consisted of mostly non-treatment-seeking individuals who were diagnosed by the Mini-International Neuropsychiatric Interview (MINI) for DSM-5 [40]. The severity of the AUD group was equally divided between mild (*n* = 5) and moderate (*n* = 3) or severe (*n* = 2). Previous research has found a positive association between drinking levels and brain and blood iron loading [22]. Thus, we expect that evaluation of a more severe AUD group would show greater accumulation of iron.

Our study found evidence for sex differences in serum and brain iron levels: women demonstrated significantly lower levels of serum iron (NIH sample) and lower brain susceptibility (Penn sample) than men. These findings are in line with previous studies reporting lower serum and brain iron loading in women than men at all life stages [41,42]. Testosterone decreases hepcidin expression, which may result in elevated iron loading in men [43]. Moreover, menstrual blood reduces serum iron levels in women, which may additionally contribute to sex differences in brain iron accumulation in pre-menopausal women [44].

Our study has several limitations. Foremost among these is the small sample size of the Penn sample, which may explain the failure to detect whole-brain and region-specific differences in iron concentration between AUD individuals and healthy controls. Although our findings of age-related brain iron accumulation are consistent with previous QSM studies, a wider age range could provide greater clarity regarding the interaction between age and iron accumulation. Nevertheless, our pilot study is unique in that it measured four-echo three-dimensional images of tissue magnetic susceptibility, which allow for more accurate and precise measurement of brain iron loading than the MRI techniques used in previous QSM studies in AUD participants [6,22]. Further, our study did not include genetic variation of the *HFE* gene, which is the main cause of hereditary hemochromatosis [45]. The *HFE* gene mediated peripheral iron loading (% saturation) in individuals with AUD [7] and healthy volunteers [46], and menstrual status in women has been found to be protective against phenotypic expression of haemochromatosis [47]. While our QSM study was underpowered to perform a genetic analysis of *HFE*, we found that, in the NIH sample, the main effect of AUD on serum ferritin remained significant when controlling for *HFE* rs1799945 (see details of the genetic analysis in Appendix A). Moreover, our screening process excluded participants with iron-related disorders such as anemia and hemophilia. Therefore, our findings of AUD on serum ferritin and brain iron loading were unlikely to be confounded by *HFE* rs1799945. Additional studies are needed to explore the use of QSM in combination with other neuroimaging techniques, such as diffusion tensor imaging (DTI), fMRI [48], and PET measures of brain glucose metabolism, which would provide a more comprehensive picture of brain structure and function in relation to AUD and aging. Assessment of neurocognitive and motor functioning is needed to better understand the clinical implications of iron accumulation in the brain in AUD. Another limitation is the lack of an assessment of nutritional status of participants, as differences in dietary iron intake may affect serum and brain levels of ferritin and iron [17]. Moreover, individuals with AUD could have altered micronutrient intake and absorption [49], which could have confounded our results. However, a nutrient-poor diet in conjunction with impaired absorption would be expected to reduce iron levels, which was not observed in the present study. Future study on serum and brain iron should aim to correct for nutritional status and dietary iron levels. Another direction for future study is the investigation of cognitive performance in relation to region-specific brain iron deposition in individuals with AUD. A systematic review of regional associations between brain iron and cognition [50] showed that iron accumulation in the globus pallidus correlated significantly with general cognition/IQ, motor function, and reaction times. Moreover, pallida iron levels correlated with memory in one study [51], but not in a second study [37].

In sum, our pilot study provided evidence for elevated serum and brain iron loading in AUD and aging. However, studies of larger samples are required to determine the potential clinical applications of QSM in the diagnosis and treatment of AUD and age-related neurodegenerative disorders. A fuller understanding of the interaction of chronic heavy drinking and aging on iron accumulation in brain could provide a basis for the development of iron-mediated therapies (such as iron chelators) to mitigate potential neurodegeneration in AUD.

## 4. Materials and Methods

### 4.1. Participants

#### 4.1.1. NIH Sample

A total of 1053 participants (583 individuals with AUD; 470 healthy non-dependent controls) who completed a natural history protocol at the NIH Clinical Center (Unit and Clinic Evaluation, Screening, Assessment, and Management, ClinicalTrials.gov Identifier: NCT02231840) between September 2011 and March 2020, and for whom serum ferritin levels were available, were included in the study. Clinical and demographic characteristics of the sample have been published previously [7]. AUD participants were admitted to the NIH Clinical Center inpatient unit for detoxification. Withdrawal symptoms were monitored using the Clinical Institute Withdrawal Assessment for Alcohol—Revised (CIWA-Ar) [52] and, if severe, patients were treated with a benzodiazepine by medical staff. All participants provided written informed consent and the study protocol was approved by the Institutional Review Board at the National Institutes of Health (protocol number 14-AA-0181).

#### 4.1.2. Penn Sample

Study participants were recruited for study protocol NCT04616781 or NCT05015881 via online advertisements (BuildClinical, iConnect) and underwent the following screening assessment: Mini International Neuropsychiatric Interview (MINI) for DSM-5 diagnosis for AUD and other neuropsychiatric disorders [40]; 30-day Timeline Follow-back (TLFB) for daily drinking patterns [53]; Lifetime Drinking History (LDH) [54]; Alcohol Use Disorders Identification Test (AUDIT) for hazardous drinking behavior [55]; Fagerström Test for Nicotine Dependence [56]; and Shipley Institute of Living Scale-2 (Shipley-2) test for crystallized and fluid intelligence [57]. Inclusion criteria for the AUD group included the following: (1) meeting DSM-5 criteria for current AUD; (2) self-report of an average of at least 15 standard drinks per week in the last month; (3) self-report of having greater than a 1 y history of heavy drinking; and (4) having had a drink within 1 week of study visit. Healthy controls were included if they demonstrated an AUDIT score of <6 and consumed less than 15 standard drinks per month. Exclusion criteria for both groups included the following: (1) current DSM-5 diagnosis of a major psychiatric disorder other than alcohol dependence in the AUD group, and intake of psychoactive medication within 24 h of study participation; (2) history of seizures; (3) HIV seropositivity; (4) a history of head trauma with loss of consciousness for more than 30 min or associated skull fracture/abnormal MRI; (5) any contraindication for MRI (e.g., presence of ferromagnetic objects or claustrophobia); and (6) a current, clinically significant physical disease or abnormality detected by medical history, physical examination, or routine laboratory evaluation that can impact brain function. Medical history, physical examination, and routine laboratory results of the screen were evaluated by physicians prior to study inclusion. Ten individuals with AUD (3 female; 37.30 ± 15.72 years old) and eight matched non-dependent healthy volunteers (4 female; 32.50 ± 8.30 years old) were included in the study. Baseline demographics and alcohol-related clinical measures are presented in Table 1. The AUD cohort had a significantly higher BMI (*p* = 0.025). As anticipated, AUD participants reported greater alcohol consumption across various drinking behavior metrics than controls, including age at first drink (14.6 vs. 18.0), standard drinks per week (25.12 vs. 0.52), total lifetime alcohol intake (190,596.00 g vs. 8557.50 g), and AUDIT score (14.8 vs. 1.75) (Table 1).

### 4.2. Quantitative Susceptibility Mapping (QSM)

On the day of study screening, Penn participants completed a serum iron panel (Labcorp) after an overnight fast. On a separate study day, participants completed a 3T MRI scan with QSM to quantify brain iron concentrations non-invasively. Participants were instructed to eat a light meal 2–3 h before the MRI scanning procedures. MRI was performed on a Siemens Prisma 3T system (Siemens Healthcare, Erlangen, Germany). QSM images were acquired using a 3D multi-echo sequence with the following parameters: repetition time (TR) = 35 ms, echo time (TE) = 7.5/15/22.5/30 ms, flip angle (FA) = 15°, field of view (FOV)= 180 × 240 mm, number of slices = 96, voxel size = 1.5 × 1.5 × 1.5 mm, readout bandwidth = 200 Hz/pixel, and scan time = approximately 5 min. QSM images were reconstructed by the morphology-enabled dipole inversion algorithm zero-referenced to the ventricular cerebrospinal fluid using the MEDI toolbox [58,59]. To allow group-level analysis, QSM images were normalized to the Montreal Neurological Institute (MNI) space with 1 mm isotropic voxels and spatially smoothed with 8 mm full-width at half-maximum (FWHM) using SPM 12 (Wellcome Trust Center for Neuroimaging, London, UK).

### 4.3. Statistical Analyses

All statistical analyses were completed using SPSS (IBM, Armonk, NY, USA). Two-sample t-tests were used to compare groups on continuous variables; Chi-square tests were used to compare groups on categorical variables (sex, self-reported race, and smoking status).

We performed a linear regression analysis with ferritin as the dependent variable and AUD, age, BMI, and sex as predictors in both the NIH and Penn samples. In the NIH sample, we added C-reactive protein values, to correct for levels of inflammation.

Global brain quantitative susceptibility was calculated for each Penn participant as the average of all brain voxels in the preprocessed QSM images identified using the tissue probability maps in SPM. Linear regression was performed with global brain iron as the dependent variable with AUD, age, BMI, sex, and serum ferritin as predictors.

The preprocessed QSM images were subjected to whole-brain voxel-wise analysis to examine the following: (1) the difference in quantitative susceptibility between the AUD and HC cohorts using a two-sample t-test, while correcting for age as a covariate; and (2) the association between quantitative susceptibility and age using linear regression. Significant clusters were explored using a cluster-level FWE-corrected threshold of *p* < 0.05. Moreover, a preliminary threshold of voxel-level uncorrected *p* < 0.005 combined with cluster extent *k* > 50 was used for exploratory analyses on the effects of AUD and age on voxel-wise quantitative susceptibility. Anatomical labels of the identified regions were determined based on the Harvard-Oxford Atlas.

## Figures and Tables

**Figure 1 ijms-24-04461-f001:**
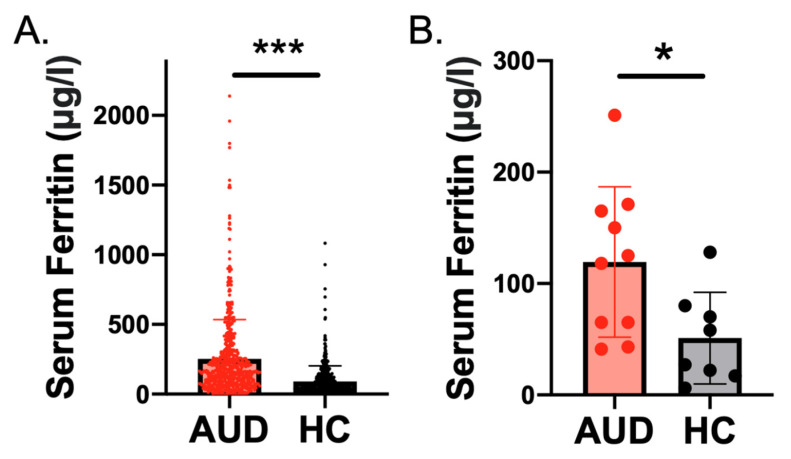
(**A**). Serum ferritin levels were higher in inpatients with alcohol use disorder (AUD; *n* = 583) than healthy controls (HCs; *n* = 470) in the sample from the National Institutes of Health (NIH) (*p* < 0.001). (**B**) Serum ferritin levels were also higher in individuals with AUD (*n* = 10) than HCs (*n* = 8) in the Penn sample (*p* < 0.05). *** *p*  <  0.001, * *p* < 0.05.

**Figure 2 ijms-24-04461-f002:**
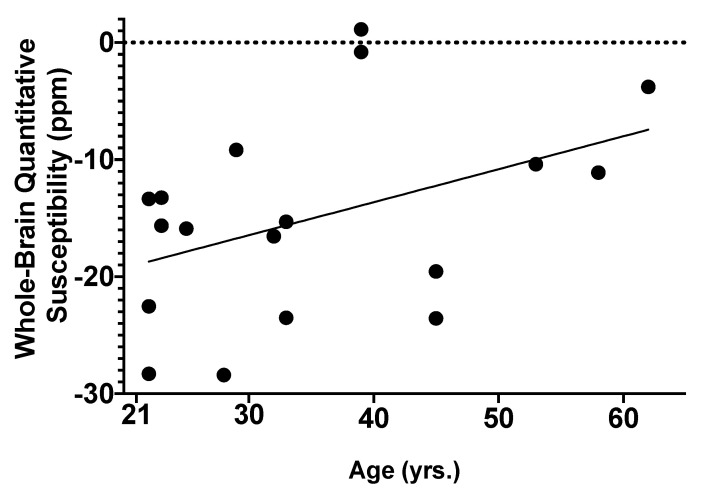
Age significantly predicted whole-brain quantitative susceptibility (*β =* 0.618, *p* = 0.020).

**Figure 3 ijms-24-04461-f003:**
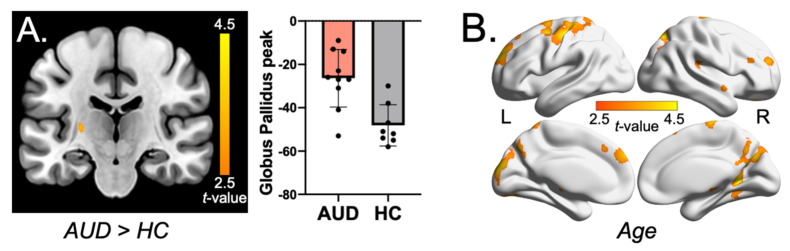
(**A**) Positive association between AUD and brain quantitative susceptibility in the left globus pallidus (peak = [−26, −21, 6], voxel-level uncorrected-*p* < 0.005, cluster extent *k* = 89). The bar graph depicts the peak susceptibility of the globus pallidus cluster for AUD > HC. (**B**) Positive association between age and brain quantitative susceptibility across participants, covering clusters in the precuneus, bilateral dorsolateral prefrontal cortex (dlPFC), orbitofrontal cortex (OFC), bilateral midbrain, bilateral fusiform gyrus/parahippocampal gyrus, left thalamus, and right caudate (voxel-level uncorrected-*p* < 0.005, *k* > 50). Abbreviations: AUD, alcohol use disorder; HC, healthy control.

**Table 1 ijms-24-04461-t001:** Demographic and clinical characteristics of the Penn alcohol use disorder and healthy control cohorts. Data are presented as mean (SD).

Characteristics	AUD (*n* = 10)	HC (*n* = 8)	*p*-Value
Age, years	37.30 (15.72)	32.50 (8.30)	*p* = 0.42
BMI (kg/m^2^)	27.51 (4.99)	22.69 (2.59)	*p* = 0.025
Education, years	15.00 (2.11)	16.75 (1.49)	*p* = 0.065
Shipley Standard IQ score	109.10 (13.71)	117.63 (12.09)	*p* = 0.19
Sex			*p* = 0.39
Female	3	4	
Male	7	4	
Self-reported race			*p* = 0.91
Black/African American	4	3	
White	6	5	
Smoking status			*p* = 0.090
Smoker	3	0	
Non-smoker	7	8	
Albumin (g/dL)	4.79 (0.21)	4.70 (0.27)	*p* = 0.44
Drinks per week	25.12 (8.65)	0.52 (0.70)	*p* < 0.001
Age of first drink, years	14.60 (2.63)	18.00 (2.58)	*p* = 0.019
Total lifetime drinks (g) (14 g per drink)	190,596.00 (332,232.50)	8,557.50 (10,546.07)	*p* < 0.001
AUDIT score	14.80 (6.11)	1.75 (1.04)	*p* < 0.001

Abbreviations: AUD, alcohol use disorder; AUDIT, Alcohol Use Disorders Identification Test; BMI, body mass index, HC, healthy control.

**Table 2 ijms-24-04461-t002:** Summary of regression analysis findings for global brain iron from the Penn sample in 10 individuals with AUD and 8 healthy controls.

	Global Brain Susceptibility (Penn)		
Variable	B	SE B	*β*
AUD	−3.55	4.58	−0.212
Age (years)	0.411	0.146	0.618 **
BMI	−0.627	0.442	−0.344
Sex	−8.375	3.789	−0.491 *
Ferritin (μg/L)	0.047	0.035	0.357
*R^2^*	0.354		
*F*	2.861 ^		

Abbreviations: AUD, alcohol use disorder; BMI, body mass index. Dummy coding: AUD (1 = AUD, 0 = healthy control); sex (1 = female, 0 = male). ^ *p* < 0.09, * *p* < 0.05, ** *p* < 0.01.

## Data Availability

Data available on request due to restrictions.

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
