# Peer review of "Peripheral and Central Iron Measures in Alcohol Use Disorder and Aging: A Quantitative Susceptibility Mapping Pilot Study"

_ijms, 2023, doi:10.3390/ijms24054461_

Round 1

Reviewer 1 Report (Previous Reviewer 1)

The authors have addressed all my concerns successfully. 

Author Response

Reviewer 2 Report (New Reviewer)

The authors search correlation between blood and CNS iron level increase and alcoholism and/or aging. They found connections between serum ferritin levels and brain iron deposition. They admit several limitations of their experiments but claim that the imaging technique used is new, and their results are not always consistent with previous findings.

I agree that the number of participants is very low, especially if we consider that those ten people gave a heterogenic group (different levels of AUD).

It is said that male persons have higher brain iron levels. How is the sex difference with age (older women and men)? With chronic hepatitis, the iron levels in the blood and liver are elevated. Is it known what happens in the brain in this case? On figure 3.A we can see the involvement of the left globus pallidus. Do you have any idea to explain the side difference (right-handedness)?

Author Response

Reviewer 3 Report (New Reviewer)

Iron is essential for all human cells. It is a cofactor of many enzymes and is involved in DNA synthesis, oxygen transport in the blood and many other physiological processes of the human body.

In their study, Adams AR et al investigated if the serum and brain iron loading is higher in individuals with alcohol use disorder (AUD) compared to healthy controls and if this process is age dependent. They observed elevated serum and brain loading in AUD and aging.

This study has some merit because of the originality of the work and the analysis of both, namely serum and brain iron loading.

The manuscript is very well written They present the manuscript in a good scientific English.

However, some point should be considered before publication:

-        - The “Introduction” is too long and should be shortened significantly.

-       -  Page 5, lines 100-102: The first sentence is not part of the “Results” and should be part of the “Introduction”.

-       -  Figure 1: AUD, HC, NIH should be spelled out in the “Figure legend”

-     -    “Discussion”: page 13, line 254: “…,which is not what was seen in the present study.” This parte of the sentence should be rewritten.

Author Response

Reviewer 4 Report (New Reviewer)

Adams, et al. present the manuscript "Peripheral and Central Iron Measures in Alcohol Use Disorder and Aging: A Quantitative Susceptibility Mapping Pilot Study."  The authors present a well-controlled and interesting study attempting to connect circulating ferritin levels to QSM values in an attempt to gain an understanding of the cognitive changes associated with AUD.  The authors validated that serum ferritin levels were higher in the AUD group than the controls, but did not associate with global brain magnetic susceptibility.   Despite being a small study, the authors do a good job of acknowledging and expanding upon study limitations. 

Some minor comments:

1.  The authors discuss the association of hepcidin and ferritin in the introduction, was circulating hepcidin measures evaluated as part of this study? 

2. While ferritin can be indicative of changes in iron homeostasis, it is also largely reflective of chronic inflammation.  Is this the case in AUD?  This should be commented on in the discussion for completeness. 

3.  It is unclear how the ROIs were established for QSM analysis across various brain regions, please clarify.  

Minor corrections:

1. Missing comma on page 1, line 25 

2. Page 2, line 33 authors use the phrase "QSM mapping" which is redundant because QSM is an acronym for "quantitative susceptibility mapping", thus, should be referred to as QSM.  There are several instances of this.   

3.  Authors describe "global brain quantitative susceptibility", the use of quantitative in this context is unnecessary and can be removed.

Round 2

Reviewer 2 Report (New Reviewer)

The authors answered all my questions correctly. I recommend the paper for publication.

This manuscript is a resubmission of an earlier submission. The following is a list of the peer review reports and author responses from that submission.

Round 1

Reviewer 1 Report

Adams et al  in their work entitled "Peripheral and Central Iron Measures in Alcohol Use Disorder: A Quantitative Susceptibility Mapping Pilot Study"  present an interesting study with impressive number of subjects that attempts to establish a connection between systemic- and brain iron levels. 

The study is mainly correlative and  lacks in depth analysis of systemic iron parameters. This reviewer is aware that the authors have previously published a study on transferrin saturation and AUD. 

i) There is no information on dietary iron levels. 

iii) The authors should discuss the  functional implication of region-specific brain iron deposition in the light of their findings. 

iv) Also,  It is not clear if the study is about both AUD and aging. The title only mentions AUD. 

Reviewer 2 Report

The manuscript describes a pilot study aimed to analyze both serum and brain iron loading in individuals with alcohol use disorder (AUD). The authors employed serum ferritin quantitation as a measure of peripheral iron content and QSM analysis to quantify brain iron. By analyzing data from 583 AUD subjects and 470 controls, they found higher serum ferritin levels in AUD individuals than in healthy subjects and by a linear regression model they demonstrated that AUD status is a predictor of serum ferritin level. They further replicate these data in a smaller cohort (Penn cohort: 10 AUD subjects and 8 healthy controls). Whole-brain iron susceptibility analysis in this small cohort did not evidence a significant association of AUD with brain iron, although Voxel-wise QSM mapping analysis revealed higher iron content in the left globus pallidus.

The study has two great limitations.

The first is the number of participants, although this limit is acknowledged by the authors in the discussion. Despite being a pilot study, 10 + 8 subjects are however really too few to draw conclusions. Furthermore, as acknowledged by the authors, a genetic study of the participants was not conducted, so the authors cannot exclude the presence of genetic variants related to iron overload conditions.

A second great limitation is that the authors quantified only serum ferritin level as a surrogate measure of peripheral iron level. However, serum ferritin is also enhanced by several other conditions like autoimmune diseases, acute and chronic inflammation or several infections (also different from HIV positivity), for example. The authors described inclusion criteria but did not consider inflammatory disorders or acute/chronic infections (or further conditions that elevate serum ferritin) as exclusion criteria.

These important limits, together with the too small number of studied subjects, make the study too preliminary to be published.       

Round 2

Reviewer 1 Report

The authors have successfully addressed the concerns. 

Reviewer 2 Report

I feel that limitations of this work are still too many.

Further, I fully disagree with the authors regarding their assertion about the clinical use of serum ferritin. Serum ferritin may increase due to a lot of clinical state so it does not reflect iron body stores.